# Characterization and Development of Gelatin from Cow Bones: Investigation of the Effect of Solvents Used for Soaking Beef Bones

**Siti Fatimah [1,2,\*], Sarto Sarto [1], Moh Fahrurrozi [1] and Budhijanto Budhijanto [1]**

[1] Department of Chemical Engineering, Faculty of Engineering, Universitas Gadjah Mada, Jl. Grafika, Yogyakarta 55281, Indonesia

[2] Department of Chemical Engineering, Faculty of Engineering, Universitas Muhammadiyah Surakarta, Jl. Pabelan, Surakarta 57169, Indonesia

\* Correspondence: sf120@ums.ac.id

**Abstract:** Beef bones contain a gelatin component that can be further extracted. This extraction process requires the right solvent to produce good yield and quality. Gelatin has multifunctional properties, namely biodegradable, biocompatible and non-toxic. This is because it is a natural ingredient that contains high amino acids. The most dominant amino acid content as a parameter for determining the quality of gelatin is proline, glycine, and hydroxyproline. The purpose of this study was to study the effect of the solvent used as a medium for soaking cow bones to be converted into gelatin. The solvent variations used include NaOH, HCl, $H_2SO_4$, $CH_3COOH$, and $NaHCO_2$. The concentration variations are 4, 5, 6, 7, 8 (%). This research method includes the preparation of cow bone samples, fat removal, mineral removal, soaking for 7 days, and extraction. The extraction process was carried out with variations times of 4 h and 6 h. The results of the study showed that the highest yield value was with 5% HCl solvent with 4 h extraction time of 26.5% with 8.67% water content, 0.9% ash content, pH 4.64, and viscosity 3.19 cP ($p < 0.05$). A good isoelectric point is produced when using an acidic solvent, which is between 5.3–5.8. The cross-linking of gelatin with chitosan, glutaraldehyde, and glucose was successfully carried out with the FTIR absorption indicator at a wavelength of 3200 cm$^{-1}$, which indicates the presence of hydrogen bonds, 1022 cm$^{-1}$, which indicates the breakdown of aldehyde bonds in glutaraldehyde compounds into C-O bonds. According to the microbial test, when gelatin is combined with chitosan, there will be an increase in the microbial inhibition zone. This shows that the development of gelatin materials is very prospective and promising.

**Keywords:** beef bones; gelatin; solvents; biodegradable; biocompatible; mechanical properties





## 1. Introduction

Bone is basically a cartilage-like connective tissue consisting of cells housed in lacunae and collagen fibers [1]. Bone is composed of one cell that is present in each lacuna and is connected to each other by a series of bones that traverse a matrix. This matrix contains collagen fibers, albuminoid substances, and calcium salts [2]. Bone is a hard tissue in the body that consists of two types of tissue, namely compact tissue and spongy tissue, which contains almost the same amount of collagen [3]. The color of fresh bone is yellowish white, and when boiled, it will become pure white. Bone consists of organic and inorganic materials, mostly inorganic materials such as calcium phosphate and calcium carbonate. At the same time, the rest are ions such as Mg, K, F, CI [4]. Inorganic materials in bone function to provide hardness to the bone structure.

Cow bone is a very promising material as a new material, especially for applications in the medical, textile, and medical textile fields [5]. This is because the content of compounds in cow bones is very compatible to be accepted by the body, for example, gelatin [6]. Gelatin is a natural product obtained from the partial hydrolysis of collagen, which is widely

found in the skin, muscles, and bones of mammals, fish bones, and chicken leg skin [7]. The main properties of gelatin are biodegradable, biocompatible, and non-toxic because it is a natural ingredient that contains high amino acids. The most dominant amino acid content as a parameter for determining the quality of gelatin is the content of proline and hydroxyproline [8]. The main component of gelatin is a protein linked by peptide bonds to form long polymer chains. Gelatin has the ability to form a gel that is reversible, easily soluble in hot water, and capable of forming a unique binding action [9]. The chemical structure of gelatin is shown in Figure 1 [10].

**Figure 1.** Structure of Gelatin.

In general, gelatin contains amino acids such as glutamic acid, aspartic acid, arginine, proline, hydroxyproline, lysine, isoleucine, methionine, leucine, and valine. Gelatin from pork and beef contains more complex amino acids [11]. The general operating conditions of gelatin synthesis are influenced by the solvent for immersion, immersion time, hydrolysis temperature, hydrolysis process time, and stirring factors. Soaking time is the time required to convert the collagen into gelatin. This is influenced by environmental conditions such as the solvent used. The solvent used to soak the bones is the key to the success of the resulting solution. If the solvent used can interact perfectly, most of the collagen will be converted into gelatin so that the yield is high.

In a study conducted by Alipal et al., the solvent used was 4% HCl [3], and the bone soaking process was for 5 days. The extraction process presented is dominated by using a biocatalyst. The use of this biocatalyst will provide additional costs, which are more expensive because most biocatalysts are relatively expensive. While the research that has been conducted by Lin et al., apart from an economic review is also a "halal" factor [12]. Some regions really consider cultural, religious, and social factors of the community. Rashedi et al. have conducted research on the use of poly(lactic acid) as an additive to improve the mechanical properties of gelatin nanofibers crosslinked with glutaraldehyde [13]. Poly(lactic acid) is relatively expensive and has toxicity when combined with glutaraldehyde.

In this study, the effect of the type of solvent used in the bone immersion process will be studied. The process of soaking beef bones can be done with an acid solvent or an alkaline solvent [14]. Different solvents used will give different performances as well. Used as a solvent, among others, are strong acids, weak acids, strong bases, weak bases, and salts of strong acids and strong bases [15]. The difference in the concentration of the solvent will affect the characteristics of the resulting gelatin [16]. Parameters that will be evaluated in this study include the characteristics of chemical and physical properties. This can be used as a recommendation for further gelatin applications, especially in the biomedical field. In addition to that, it will also be researched the development of beef bone gelatin material as an advanced material in the medical field. The treatment is an antimicrobial activity test that can inhibit microbial growth so that the gelatin material can function as an antimicrobial.

## 2. Materials and Methods

### 2.1. Materials

The materials used include beef bone in the femur, acetic acid, hydrochloric acid, sulfuric acid, sodium oxide, sodium bicarbonate, and isopropanol, which were purchased from Sigma Aldrich, Global Satria Aji, Jakarta, Indonesia. Distilled water, phosphate buffer saline, aquadest, commercial chitosan, commercial gelatin, potassium bromide, glucose, ethanol, sodium chloride, and glutaraldehyde.

### 2.2. Method

The research procedure includes cleaning beef bones, size reduction, and degreasing, which is the process of removing dirt, fat, and meat residue. To remove fat, impurities, and remaining bone meat, boiled for 120 min, then filtered. The next stage is the immersion of the sample through a hydrolysis process, followed by demineralization, namely the removal of calcium and other salts contained in beef bones. Immersion was carried out for 7 days with various solvent concentrations (4, 5, 6, 7)% used for soaking. The solvents are sodium hydroxide, hydrochloric acid, sulfuric acid, acetic acid, and sodium bicarbonate. The soaking solution is changed every 2 days. After immersion, the osein solution was obtained. The next step was to filter and wash with the flowing water until the pH solution was 6–7. The next stage is the extraction of the resulting solution to produce gelatin. The next process is the extraction of the resulting osein. The extraction process is carried out in a waterbatch. The ratio of ossein to aquadest is 1:1 ($w/v$). After that, it was extracted in an oven at a temperature of 70 °C with a time variation of 4 h and 6 h and then filtered with whatman paper. The filter results are concentrated, and the concentration process is carried out with an evaporator until the concentration becomes 25–30% with a concentration temperature of 80 °C for ±2 h. After concentration, the sample was put in a refrigerator at 40 °C until a gel was formed. The gel-shaped sample was dried using an oven at 60 °C for 48 h, and a dry gelatin sample was produced. Figure 2 shows the flowchart of the synthesis gelatin.

The next stage is the preparation of gelatin samples linked with chitosan and glutaraldehyde or glucose materials. The mixing of gelatin-chitosan was carried out in a ratio of 2:3. The mixing of gelatin-chitosan-glutaraldehyde was carried out at a ratio of 1:1:1, as well as the mixing of gelatin-chitosan-glucose. Before mixing, the chitosan was dissolved in 1% acetic acid. After completely dissolved, add glutaraldehyde or glucose. Stirred using a magnetic stirrer for 20 min at a temperature of 70 °C, then the samples were dried and analyzed.

### 2.3. Preparation of Characterization

#### 2.3.1. Determination of Yield

The yield value of gelatin extraction was calculated using the Equation (1):

$$\% \ Yield = \frac{weight \ of \ dry \ gelatin \ (g)}{weight \ of \ cow \ bones \ used \ (g)} \ \times \ 100\% \tag{1}$$

#### 2.3.2. pH

The pH of gelatin samples can be analyzed using a digital pH meter. The gelatin solution was prepared in a concentration of 1% ($w/v$) using distilled water at 60 °C, then constantly stirred for 30 min and cooled at room temperature (25 °C). The pH of the gelatin solution was measured using a pH meter [17].

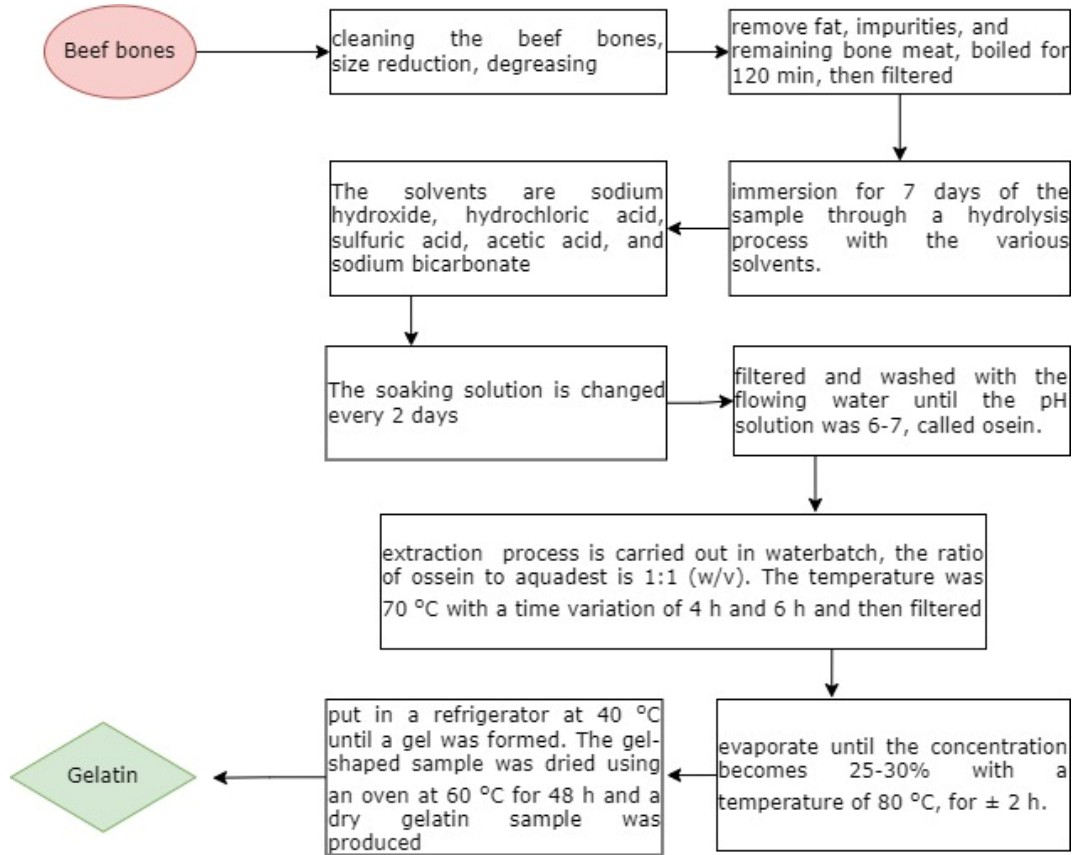

**Figure 2.** Flowchart of Synthesis Gelatin.

### 2.3.3. Determination of Viscosity

Each sample of cow bone gelatin and commercial gelatin was weighed as much as 6.67 g and then dissolved in distilled water to a volume of 100 mL. Then cooled at 20 °C and the viscosity was measured using a Brookfield digital viscometer using spindle No. 1 at 30 °C $\pm$ 0.5 °C.

### 2.3.4. Determination of Ash

Ash content testing was carried out using the method [6]. A sample of 2 g of gelatin was weighed and placed in an ashing dish that had previously been weighed and burned in a kiln at 600 °C and cooled in a desiccator. The gelatin sample was put into an ashing cup, then closed and put into an ashing furnace and burned until grayish ash was obtained. The ashing process is carried out in 2 stages, namely, the first stage at a temperature of 400 °C for 1 h and the second stage at a temperature of 550 °C for 5 h. The cup is then cooled in a desiccator and then weighed. Ash content can be calculated using the formula:

$$Ash\ (\%) = \frac{weight\ of\ ash\ (g)}{weight\ of\ gelatin\ used\ (g)} \times 100\% \tag{2}$$

### 2.3.5. Moisture

The water content test was carried out using the method [18]. A sample of 2 g of gelatin was weighed and placed in an empty cup that had previously been weighed. The cup and lid were dried in the oven and cooled in a desiccator. The gelatin sample was put into a cup, then closed and put in an oven at a temperature of 100–102 °C for 6 h. The

cup is then cooled in a desiccator and then weighed. The water content is calculated using the formula

$$Moisture \ (\%) = \frac{W_{before \ drying} - W_{after \ drying}}{weight \ of \ gelatin \ used \ (g)} \times 100\% \qquad (3)$$

### 2.3.6. Point of Isoelectric

The isoelectric point test was carried out qualitatively, using a 3.8 acetate buffer; 4.7; 5; 5.3; 5,9. The resulting gelatin was taken as much as 1 g and dissolved in 10 mL of distilled water, then added a buffer solution of 2 mL each. The solution was shaken and heated in a water bath; changes were observed at 0, 10, and 30 min. The time at which precipitate or turbidity is formed is the isoelectric point of gelatin. Formation of precipitate is represented by a number, number (1) indicates no precipitate, and the solution is not cloudy, number (2) indicates a cloudy solution and no precipitate, and number (3) indicates a solution-producing precipitate and turbid.

### 2.3.7. Microbial Test

Microbial testing was carried out using Escheria coli bacteria. Testing for gelatin was carried out using the APM method, namely by counting the number of microbes using a liquid medium in a test tube. For testing gelatin which has been combined with chitosan, glucose, and glutaraldehyde, is carried out using the Well Diffusion method (wells) [19].

### 2.3.8. FTIR Spectroscopy

Sample preparation was carried out by cutting the gelatin sample with a size of $2 \times 2$ cm, then pelleted with KBr, and scanned at 25 °C. The variation of main groups of the gelatin molecule with different solutions was characterized by FTIR spectroscopy Shimadzu, IRSpirit, from Kyoto, Japan. Measurements were performed at room temperature and relative humidity of less than 40%. The scan processing range was $4000-500$ cm$^{-1}$ with $4$ cm$^{-1}$ resolution.

### 2.3.9. Scanning Electron Microscopy (SEM)

The morphology of the film was observed by Scanning Electron Microscopy (SEM), JEOL-JCM 7000, from Kyoto, Japan. The gelatin samples were sprayed with gold with a sputter coat before testing. The surface and cross-section morphologies of the samples were visualized at 20 kV.

### 2.3.10. Statistical Analysis

Statistical tests were carried out using SPSS 23.0 series software. A completely randomized design was performed to represent mean ± standard deviation (SD) values. One-way Analysis of Variance (ANOVA) was conducted, and Duncan's Multiple Range Test (DMRT) was done for mean comparison. SPSS package (SPSS 23.0 for Windows, SPSS Inc, Chicago, IL, USA) software was used for data analysis [19].

## 3. Results and Discussion

The synthesis of the gelatin stage starts with the preparation of beef bones, washing, boiling (degreasing), followed by the demineralization process and extraction. The purpose of this stage is to obtain gelatin material in accordance with the Indonesian National Standard (SNI 01-3735-1995). At this stage, good operating conditions are sought so that the desired material is produced. These operating conditions include the selection of solvents used for soaking beef bones, namely acidic and alkaline solvents. Acid solvents used include HCl, $H_2SO_4$, and $CH_3COOH$, while the alkaline solvent used is NaOH. Salt solvents are also carried out, namely by using $NaHCO_2$. Other variables were the length of time for bone immersion, the concentration of the solvent used to soak the bones, and the extraction time of gelatin. After obtaining the material, characterization was carried out,

namely physical tests, characterization of functional groups by FTIR, and morphological analysis by SEM-EDX.

The manufacture of gelatin in this study uses beef thigh bone or fermur, where the bone is more easily separated from the meat and the fat attached. Immersion treatment using NaOH solution so that the resulting gelatin is a basic type of gelatin. Immersion was carried out for 7 days, with the replacement of the solution every two days to prevent changes in concentration. The difference in concentration of NaOH solution greatly affects the bone structure of cows. The higher the NaOH concentration, the more porous and eroded beef bones were compared to a lower concentration where the beef bones were still intact and a little hard. Analysis to determine the characteristics of this gelatin is an analysis of yield, water content, ash content, pH, viscosity, and microbiological tests.

The statistical analysis is shown in Table 1. Table 1 shows that the yield values range from 10.64% to 19.04%. Table 1 explains that soaking with different solvents and different extraction process times has significantly different effects ($p < 0.05$). The difference in extraction time, which is 4 h and 6 h, has a different effect on the resulting yield value.

**Table 1.** Values of Mean $\pm$ Standard Deviation.

| Sample | Yield (%) | Mouisture (%) | Ash (%) | pH | Viscosity (cP) |
|---|---|---|---|---|---|
| NaOH, 4 h | $11.13 \pm 1.16$ | $7.13 \pm 1.68$ | $0.60 \pm 0.15$ | $6.15 \pm 0.04$ | $2.51 \pm 0.15$ |
| NaOH, 6 h | $12.61 \pm 2.76$ | $6.06 \pm 3.03$ | $0.65 \pm 0.12$ | $6.07 \pm 0.06$ | $2.61 \pm 0.15$ |
| HCl, 4 h | $19.04 \pm 5.30$ | $5.82 \pm 1.32$ | $0.77 \pm 0.15$ | $4.41 \pm 0.66$ | $3.23 \pm 0.26$ |
| HCl, 6 h | $19.96 \pm 5.86$ | $3.94 \pm 2.34$ | $0.83 \pm 0.18$ | $4.55 \pm 0.13$ | $3.14 \pm 0.16$ |
| $H_2SO_4$, 4 h | $17.14 \pm 5.30$ | $8.67 \pm 1.48$ | $0.87 \pm 0.15$ | $4.15 \pm 0.66$ | $5.23 \pm 0.26$ |
| $H_2SO_4$, 6 h | $18.96 \pm 5.86$ | $6.60 \pm 3.09$ | $0.93 \pm 0.18$ | $4.30 \pm 0.31$ | $5.14 \pm 0.15$ |
| $CH_3COOH$, 4 h | $10.14 \pm 5.30$ | $9.67 \pm 1.48$ | $0.94 \pm 0.15$ | $2.30 \pm 0.55$ | $5.73 \pm 0.26$ |
| $CH_3COOH$, 6 h | $11.96 \pm 5.89$ | $7.60 \pm 3.08$ | $1.01 \pm 0.18$ | $2.05 \pm 0.31$ | $5.64 \pm 0.16$ |
| $NaHCO_2$, 4 h | $10.64 \pm 5.30$ | $12.17 \pm 1.48$ | $1.02 \pm 0.15$ | $9.4 \pm 0.66$ | $8.23 \pm 0.26$ |
| $NaHCO_2$, 6 h | $12.46 \pm 5.86$ | $10.60 \pm 3.08$ | $1.08 \pm 0.18$ | $9.56 \pm 0.31$ | $8.14 \pm 0.16$ |

## 3.1. Yield

Yield results are very important in determining the efficiency of the treatment applied in the manufacture of gelatin without neglecting other properties. The increase in yield value can be attributed to how much collagen is converted to gelatin. The use of strong bases or strong acids in the immersion/demineralization process can increase the presence of hydroxyl $OH^-$ ions or $H^+$ ions and accelerate the hydrolysis process. The faster the increase in the conversion of collagen to gelatin, the higher the yield obtained. Based on Table 1, it can be seen the effect of soaking NaOH with different concentrations and extraction times on the yield of beef bone gelatin. At the time of the extraction for 4 h, bone immersion with 4% NaOH produces gelatin with a yield of 10.391%. Immersion with 5% NaOH yield increased by 13.081%. The yield of gelatin with 6% NaOH immersion was 10.628%. This value seemed to decrease from the previous concentration. However, the immersion with 6% and 7% NaOH showed a decrease with a yield of 10.628% and 9.994%, respectively. The yield of cow bone gelatin at the time of the extraction for 6 h with 4% NaOH immersion was 11.141%. In immersion with 5% NaOH, the yield is 10.587%. This result shows a decrease from the previous concentration. On immersion with NaOH, 6% and 7% resulted in yields of 12.886% and 11.139%, respectively. Immersion with a concentration of 7% and 8% showed a significant increase from other concentrations. From these data, the effect of bone immersion with different concentrations of NaOH indicates a change in the yield ($p < 0.05$). The yield value seems to increase and decrease but tends to increase as the concentration of NaOH solution increases. This yield value is influenced by the washing process. With a high concentration of NaOH solution, some of the bone turns into bone powder and dissolves during the washing process using water. Ossein is also dissolved and wasted in this washing process which affects the reduced yield value of gelatin. However, changes in the shape of bone into powder or small flakes also affect

the increase in yield where the bone dissolved in aquadest during extraction is not filtered well and is mixed with gelatin, allowing other minerals to be added to the yield value. Collagen that has undergone acid or alkaline soaking can dissolve in water, with this washing process with water has a relatively small effectiveness. The effect of extraction time on the yield of beef bone gelatin tends to increase. The yield with an extraction time of 6 h tends to be higher than the yield of gelatin with an extraction of 4 h. The highest yield value in 4 h extraction was 13,081% by immersion with 5% NaOH, while the yield at 6 h extraction was 17.294% immersion with 8% NaOH. This high yield value indicates the optimum concentration of NaOH immersion and optimum extraction time in the process of making gelatin using alkaline solvents.

If using an acid solvent, the yield analysis, as shown in Figure 3, shows that there is an effect of soaking HCl with concentration and extraction time on the yield of cow bone gelatin ($p < 0.05$). At the time of the extraction for 4 h, the HCl immersion resulted in the highest yield of bovine bone gelatin at 5% HCl concentration, which was 26.476%, and the relative yield decreased after that as the concentration increased. While the yield of cow bone at the extraction time for 6 h, the highest results were obtained at a 5% concentration of HCl immersion, namely 25.72%, but at other concentrations of HCl immersion, fluctuating results were obtained. The high yield is related to the amount of collagen that is converted to gelatin. The use of strong acids causes an increase in the concentration of $H^+$ ions in the curing solution, which accelerates the hydrolysis process.

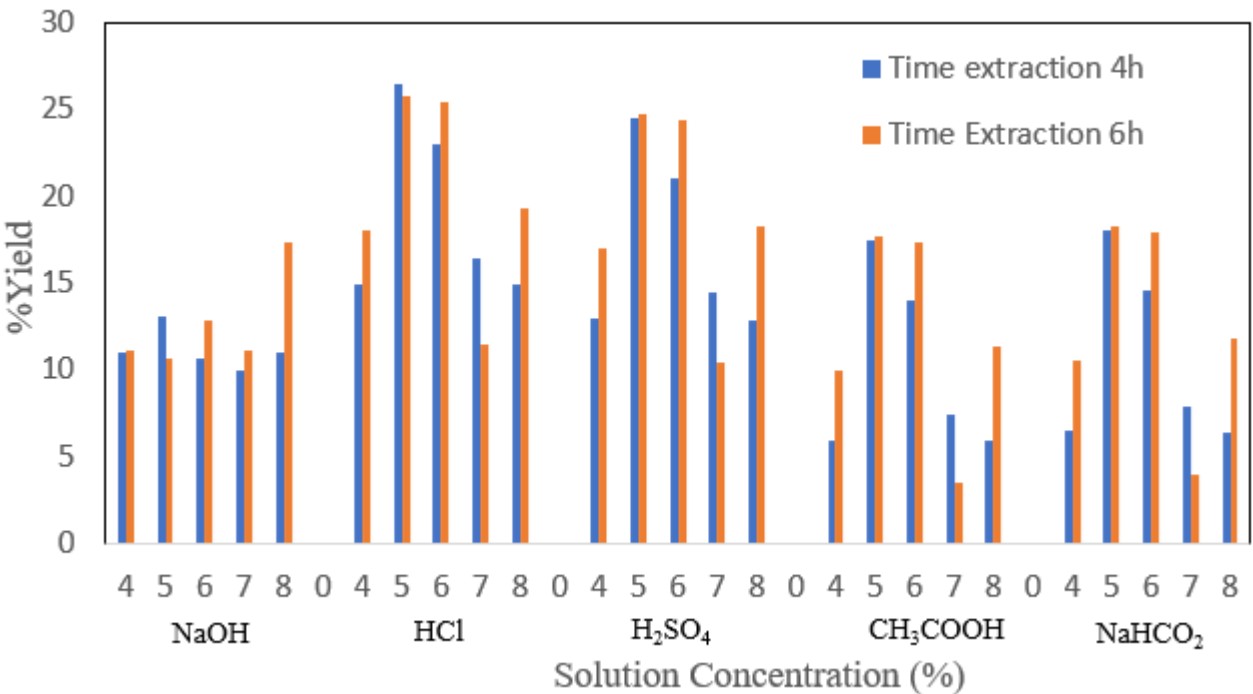

**Figure 3.** The yield of Gelatin.

The faster hydrolysis rate tends to increase the conversion of collagen to gelatin so that it can increase the yield value. Based on the results obtained, the extraction time has a tendency to increase the yield of beef cow gelatin. The longer the extraction time, the higher the yield. This is presumably because the number of $H^+$ ions that hydrolyze collagen is greater, while the extraction time causes the collagen to decompose more into gelatin. However, the very high extraction time and excessive acid concentration are thought to cause further hydrolysis of the collagen, which has been converted to gelatin, so that the gelatin becomes damaged and the yield decreases [20]. Collagen hydrolysis steps must be carried out correctly (time and concentration). Otherwise, the collagen will dissolve completely in the increasing use of HCl and increasing the immersion time, causing the

lower yield of gelatin produced because hydrochloric acid is a strong acid. The higher the use of hydrochloric acid, the higher the acid content in the solution so that it not only hydrolyzes the triple helix structure of collagen into an irregular structure but can hydrolyze peptides up to the amino acid chain so that the gelatin extraction yield becomes less [21]. Soaking too long and too strong the concentration of the soaking agent can cause collagen to be degraded and completely destroyed so that gelatin cannot be produced by solvent, causing a decrease in the yield of gelatin produced [22]

### 3.2. Water Content

The value of water content in foodstuffs has an important role that can affect the texture and determine the shelf life against damage by microbes. According to Figure 4, the increase in concentration and extraction time affects the reduction in gelatin water content. The immersion treatment with different extraction times and solvents showed different water content values. The treatment at an extraction time of 4 h produced a water content relatively higher than 6 h. The extraction time affects the water content of gelatin. The water content of the extracted gelatin for 6 h tends to be smaller. This is because at a longer extraction time, there is a longer contact and heating of the ossein, and it breaks down the guanidine and arginine groups which reduce the hygroscopic properties of gelatin [19]. The use of acid for soaking is considered to provide good performance as well.

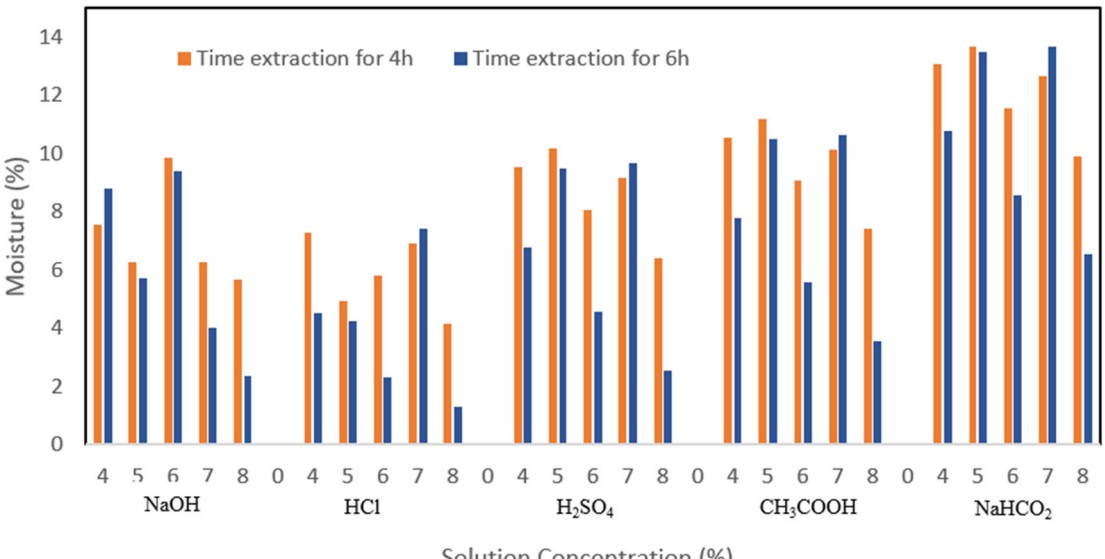

**Figure 4.** The moisture of Gelatin.

### 3.3. Ash

Ash content is one of the parameters that is used to measure the purity of a material, shown in Figure 5. The mineral content of a material can affect the high or low value of the ash content of the material. The ash content of gelatin produced from this research is relatively high. The high ash content can be caused by the demineralization process that is not optimal. The highest ash content was 1.319% with 8% $NaHCO_2$ immersion treatment with 6 h extraction time. At the same time, the lowest ash content was 1.031% with 4% NaOH immersion treatment 4 h extraction time. High ash content is also influenced by solvent concentration, where the higher the concentration of gelatin ash content tends to increase. This can also be due to less than optimal filtration, which results in other minerals still being mixed in the gelatin solution before the drying process. The extraction time did not seem to significantly affect the ash content of the gelatin produced, where the ash content of gelatin at 4 h of extraction was not much different from the ash content at 6 h of extraction. However, at 6 h of extraction, the ash content was slightly lower.

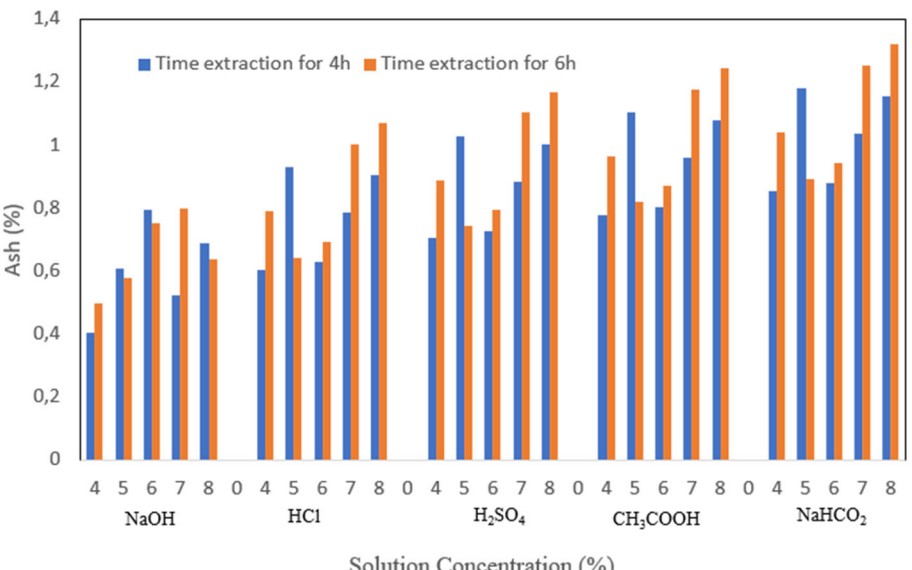

**Figure 5.** Ash Content.

*3.4. pH*

The application of gelatin in the product is affected by pH shown in Figure 6. Gelatin with neutral pH is applied to meat products, pharmaceuticals, chromatography, and paints. Meanwhile, gelatin with a low pH value is used for the food industry, and high pH gelatin is used for the pharmaceutical industry. The pH value will depend on the solvent used during the demineralization process [23]. In this study, the solvent used was acid and alkaline. The washing process played an important role in neutralizing the pH of the gelatin. For this study, washing was carried out with plain water, and this method was less effective, so it should be washed using acid so that the pH of alkaline gelatin could become more neutral. The optimum pH value in this study was the lowest pH of 10.49 in the immersion treatment with a 5% $NaHCO_2$ extraction time of 6 h. For immersion with HCl, it has a relatively acidic pH. This pH condition will affect the isoelectric point produced. Based on research conducted by Rather that the soaking process in acidic conditions will produce more gelatin [24]. At very low pH, a deprotonation process occurs and inhibits the process of forming hydrogen bonds [25]. From an economic point of view, the use of acidic solvents is cheaper [26]. For 100 mL, the price of 1 N HCl is lower than $CH_3COOH$ and $H_2SO_4$.

*3.5. Viscosity*

Viscosity testing on gelatin serves to determine the level of viscosity of gelatin as a solution at a certain concentration. From Figure 7, the effect of concentration on the viscosity of bovine bone gelatin tends to decrease. Viscosity shows that $NaHCO_2$ is not very good at hydrolyzing cow bone collagen peptides so the molecular weight of gelatin is also not so high that the viscosity of gelatin is still low [27]. The use of HCl and NaOH gives a better viscosity value than using a base. The higher the molecular weight, the slower the distribution process. The extraction time showed a slight increase in the viscosity value, where the gelatin viscosity at 4 h extraction, on average, was higher than the gelatin viscosity at 6 h extraction. The optimum viscosity in this study is the viscosity that is closest to the standard or the highest viscosity value, namely the result of gelatin treatment with acid soaking.

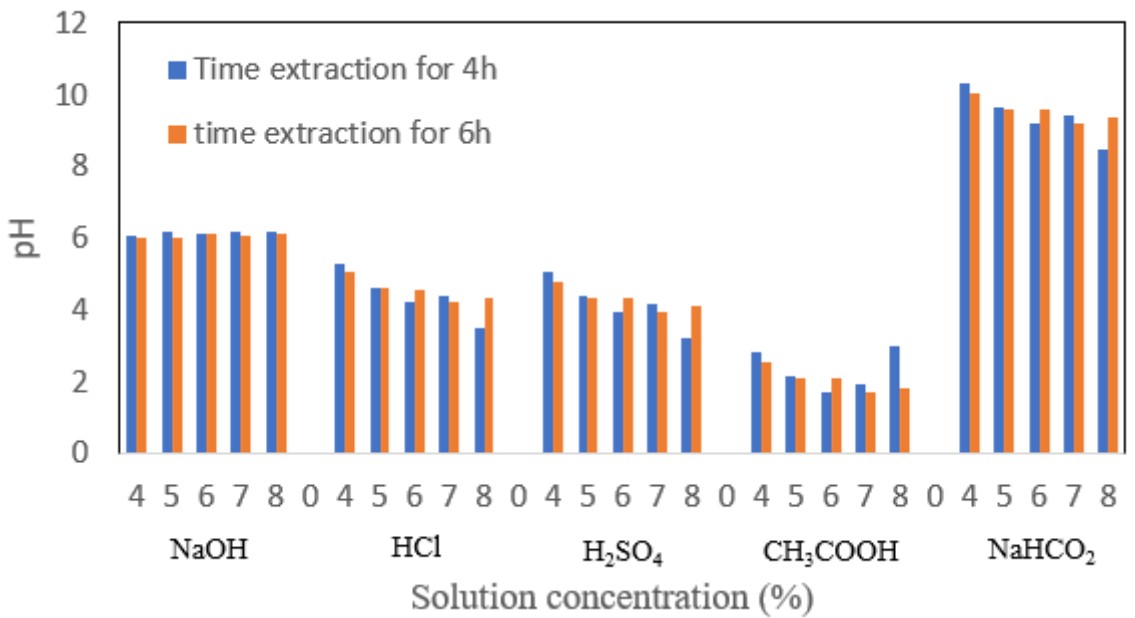

**Figure 6.** pH Value.

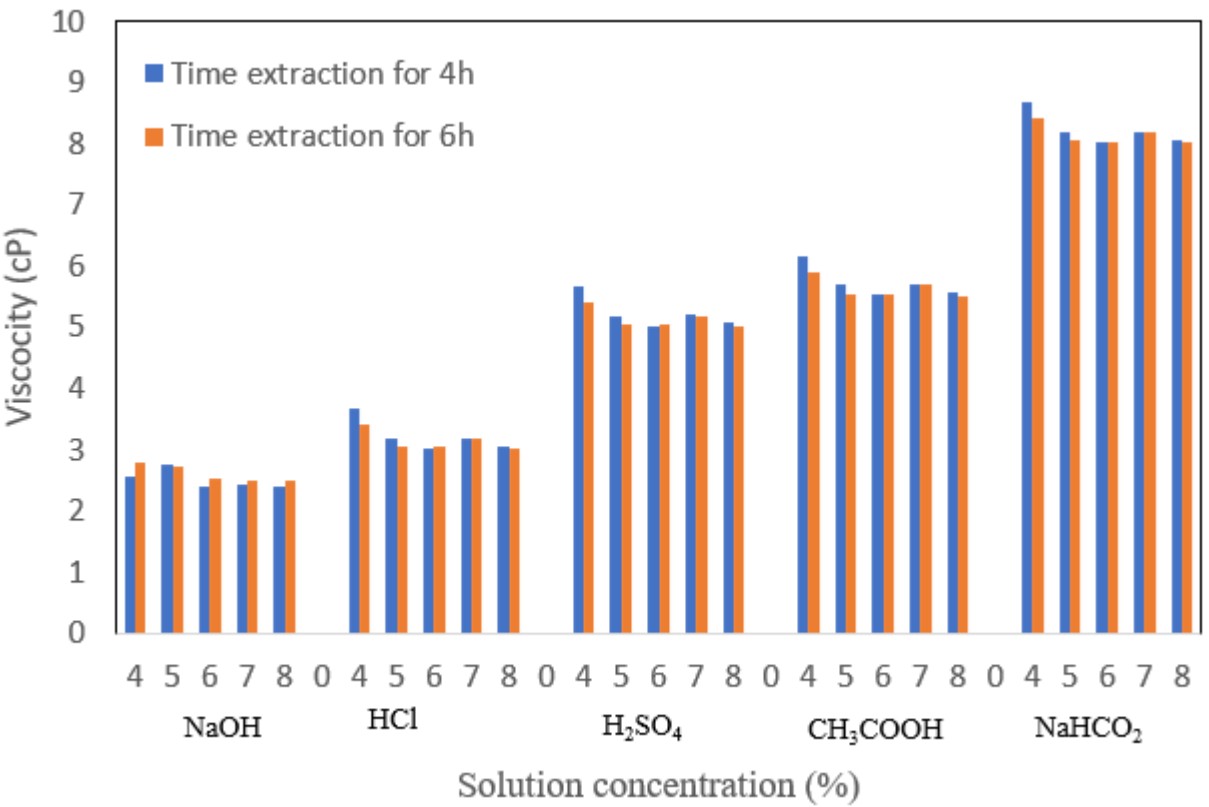

**Figure 7.** Viscosity.

*3.6. Microbial Test*

Gelatin is an excellent nutrient for some microorganisms. Therefore care must be taken during the gelatin manufacturing process to prevent possible contamination. Indonesia has determined the microbial contamination of APM (Most Likely Number). *Escherichia coli* in foodstuffs is <3/gr. Microbial contamination determines microbial development in gelatin. In this study, Escherichia coli contamination was tested on the gelatin yield, which was considered the most optimum, namely the gelatin with 6% NaOH immersion

treatment with an extraction time of 6 h with the largest yield. The results of the gelatin microbial contamination test in this study were in accordance with the SNI standard, namely <3 APM/gr, shown in Table 2.

**Table 2.** Gelatin Microbial Test Results.

| No. | Solvent | Result |
|---|---|---|
| 1. | NaOH | <3 APM/gr |
| 2. | HCl | <3 APM/gr |
| 3. | $H_2SO_4$ | <3 APM/gr |
| 4. | $CH_3COOH$ | <3 APM/gr |
| 5. | $NaHCO_2$ | <3 APM/gr |

### 3.7. Point of Isoelectric

Isoelectric point testing is done qualitatively. Based on Figure 8, precipitation was dominated at pH 5.3–5.8 during the observation time of 30 min. When compared with commercial gelatin, gelatin produced from various types of solvents has almost the same isoelectric point area. For commercial gelatin, a precipitate appears after the reaction has been running for 30 min, whereas for gelatin with various types of solvents, a precipitate occurs after the reaction has been running for 10–30 min. This shows that the isoelectric point with various types of solvents at the time of immersion of beef bones has the same value ($p < 0.05$). The types of proteins found in gelatin did not change in their peptide bonds, meaning that the composition of positively and negatively charged proteins did not change significantly. At that pH point, the protein from gelatin is easy to precipitate.

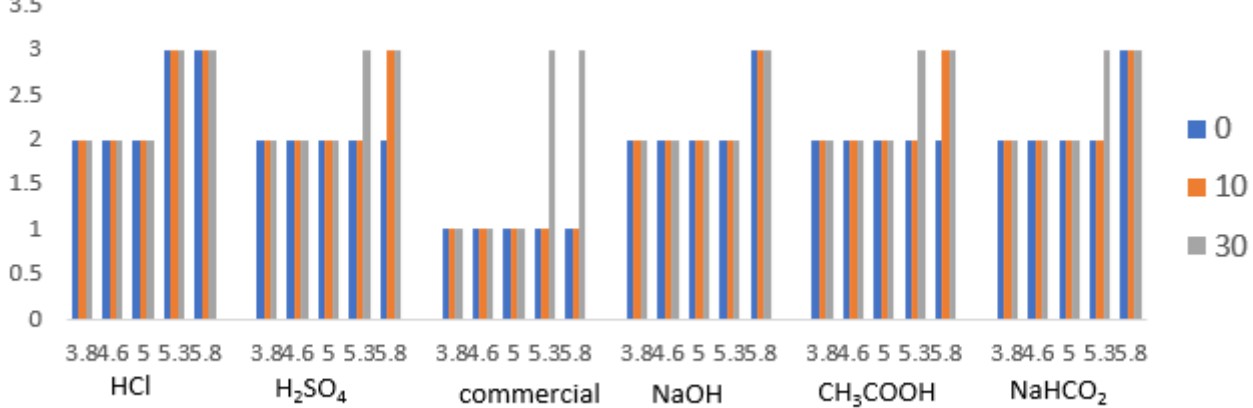

**Figure 8.** Point of Isoelectric.

### 3.8. Characterization of FTIR

Characterization using FTIR is shown in Figure 9. The FTIR results show that by using acid or alkaline solvents, there is a sharp absorption at a wavelength of about 1600 cm$^{-1}$ which indicates the C = C skeletal vibration of collagen compounds found in bone [28]. The test results obtained indicate that there is a decrease in the absorbance value at the wave number, which qualitatively indicates a decrease in the collagen content along with the increase in the concentration of osein solution as a result of the increasing number of OH- which causes swelling and dissolution of collagen by the solvent. At a wavelength of about 2800–2900 cm$^{-1}$ is a C-H vibration of the methyl group. At 1500–1600 cm$^{-1}$ wavelength, there are C-H vibrations that indicate the absorption characteristics of gelatin [29].

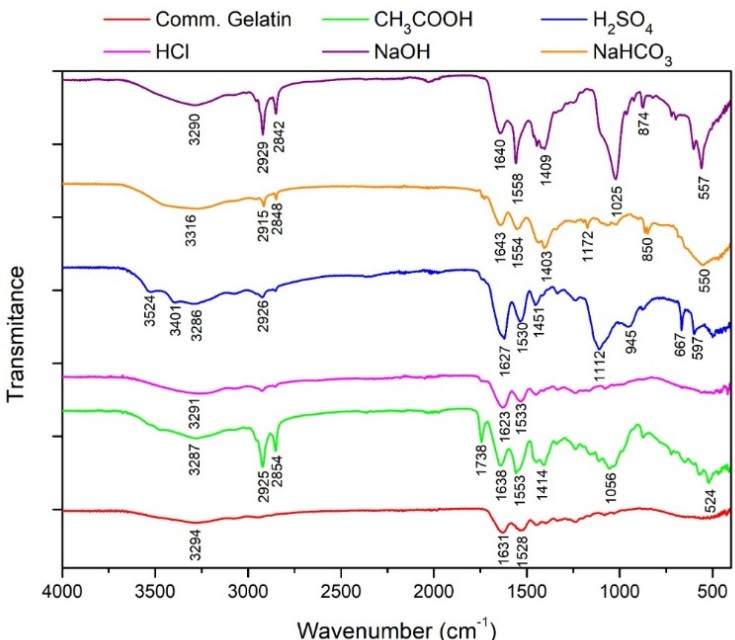

**Figure 9.** FTIR for Gelatin in Various Solvents.

At the stage of the gelatin fixation process with chitosan and also a crosslinker agent, the results of the FTIR and SEM-EDX test are shown in Figure 10 and Table 3. This cross-linking process occurs due to the formation of Schiff bases [30]. This event causes the replacement of the C = O groups from ketones or aldehydes from gelatin to be replaced by C = N–R groups from chitosan catalyzed by acid [31].

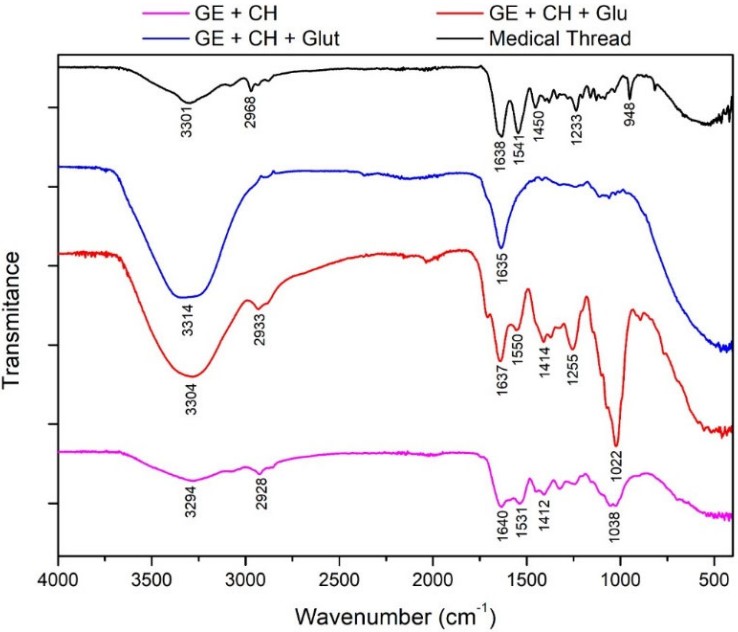

**Figure 10.** FTIR for Crosslinking Gelatin With Chitosan, Glukosa, and Glutaraldehyde.

**Table 3.** Analysis of SEM-EDX.

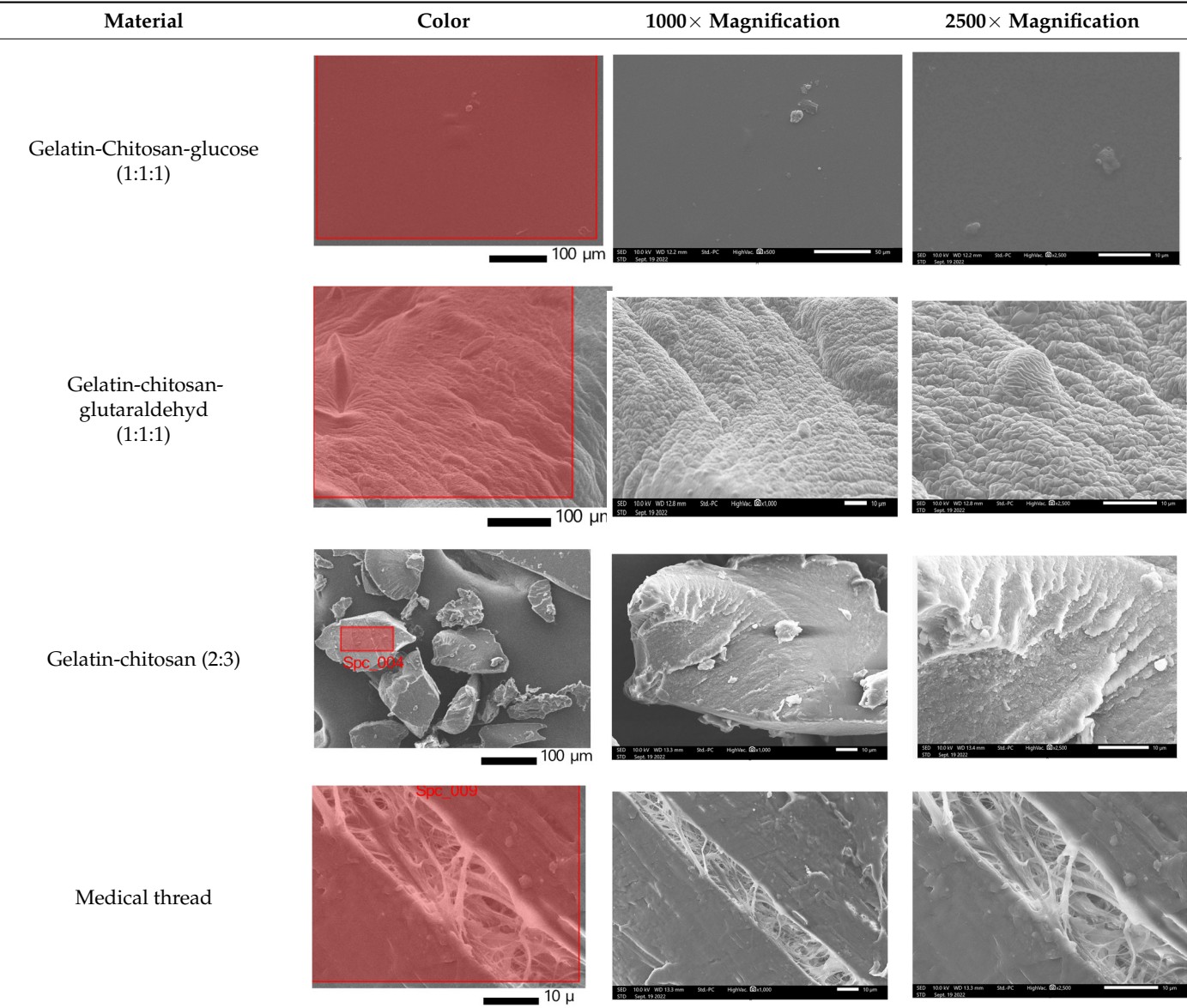

| Material | Color | 1000× Magnification | 2500× Magnification |
|---|---|---|---|
| Gelatin-Chitosan-glucose (1:1:1) | | | |
| Gelatin-chitosan-glutaraldehyd (1:1:1) | | | |
| Gelatin-chitosan (2:3) | | | |
| Medical thread | | | |

### 3.9. SEM-EDX Analysis

Based on the SEM-EDX analysis, it is suspected that the gelatin fixation process with chitosan, glucose, and glutaraldehyde crosslinker agents has occurred. Table 3 shows that the crosslinking process between gelatin-chitosan-glucose produces a smoother surface. For gelatin-chitosan-glutaraldehyde, it produces a rather porous surface. This is similar to a study conducted by Wu et al. in which gelatin-chitosan crosslinking resulted in a slightly smoother surface [32]. However, this still needs to be studied further regarding the optimization of the operating conditions carried out. The cross-linking process between gelatin and chitosan has been successfully carried out. The use of a glutaraldehyde crosslinking agent is a cheap material [12,33]. The process of crosslinking gelatin and chitosan has also been successfully carried out by Zheng et al. [34]. The gelatin-chitosan cross-linking process can increase microbial inhibition and physical strength [35]. According to the research by Colobatiu et al., the combination of gelatin-chitosan can produce health products that have the potential to prevent inflammation and infection [36]. The combination of gelatin-chitosan-glutaraldehyde produces a material whose conditions are similar to the research conducted by Tagrida et al. [19]. This material can be engineered by varying the

composition of gelatin-chitosan-glutaraldehyde/glucose, in terms of density, porosity, and surface area [37]. These characteristics hold promise for application in various fields, such as health. © EDX Analysis is shown in Figure 11 and Table 4. Ini menjelaskan bahwa komposisi atom C dan atom O beranekaragam. Hal ini diduga dipengaruhi oleh.

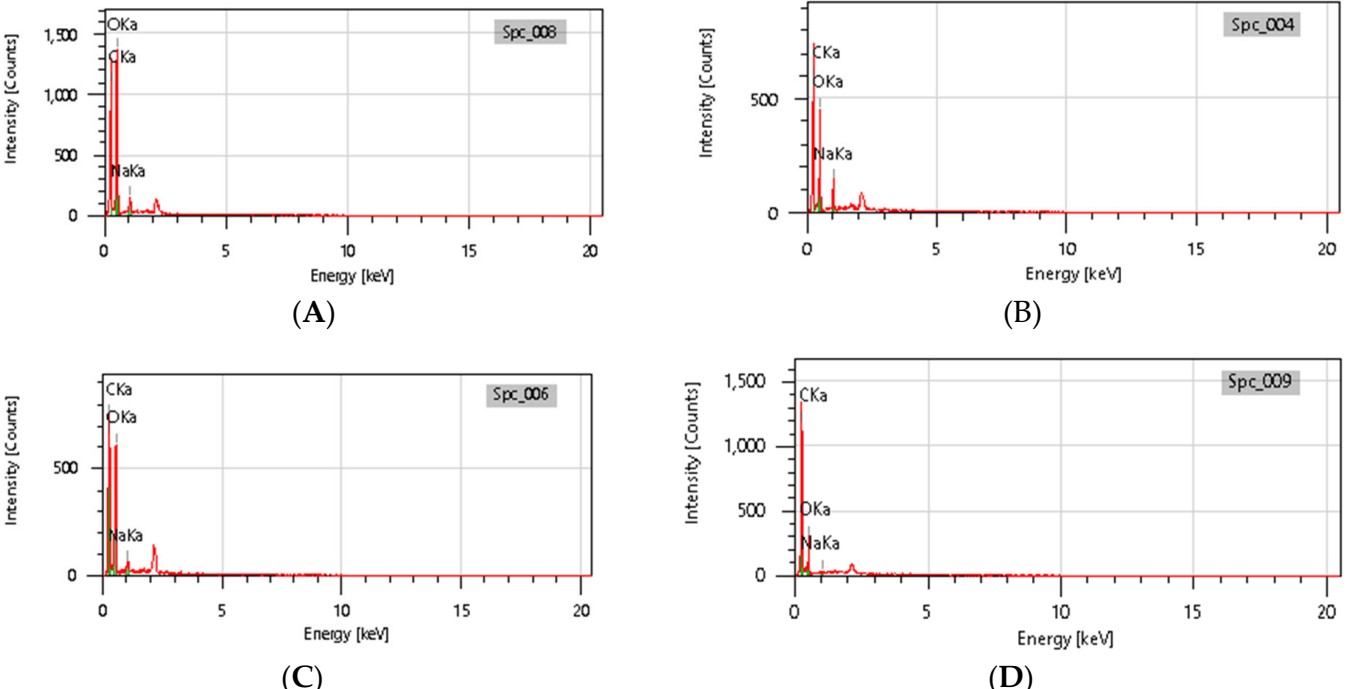

**Figure 11.** EDX Representation, (**A**) Gelatin-Chitosan-glucose (1:1:1), (**B**) Gelatin-chitosan-glutaraldehyde (1:1:1), (**C**) Gelatin-chitosan (2:3), (**D**) Medical thread.

**Table 4.** Distribution of Element C and O.

| Material | Element | | | |
| --- | --- | --- | --- | --- |
| | %Mass C | %Mass O | %Mass Na | Total |
| Gelatin-Chitosan-glucose (1:1:1) | 36.26 ± 0.67 | 60.74 ± 1.69 | 3.00 ± 0.36 | 100.00 |
| Gelatin-chitosan-glutaraldehyd (1:1:1) | 40.73 ± 0.99 | 56.83 ± 2.40 | 2.44 ± 0.46 | 100.00 |
| Gelatin-chitosan (2:3) | 42.91 ± 1.10 | 48.91 ± 2.38 | 8.19 ± 0.84 | 100.00 |
| Medical thread | 60.58 ± 1.10 | 38.99 ± 2.24 | 0.44 ± 0.20 | 100.00 |

### 3.10. Antimicrobial Test of Material

The products resulting from this study were also investigated for their anti-microbial activity. Based on the research, it was found that there is a good inhibition zone in gelatin combined with chitosan [38]. Based on Figure 12, the addition of chitosan concentration will increase the area of the inhibition zone ($p < 0.05$). The same thing has also been done in research that has been done by Islam et al. [39].

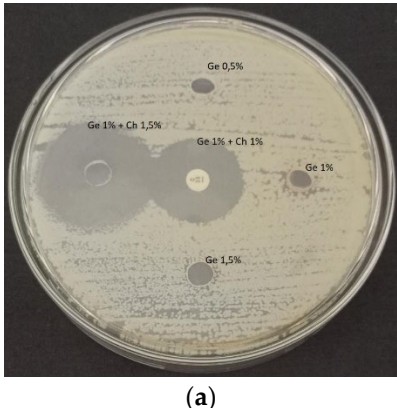
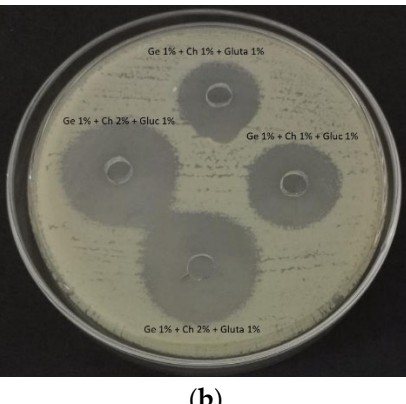

(**a**)            (**b**)

**Figure 12.** Antimicrobial Test of Gelatin Combined Material. (**a**) inhibition zone for gelatin and gelatin combined with chitosan with various concentrations, (**b**) inhibition zone for gelatin combined with chitosan and crosslinker with various concentrations.

## 4. Conclusions

The process of synthesis and characterization of gelatin from beef bones has been carried out well. Based on research, the type of solvent will have a very significant effect. Solvents that will produce good yields and physical characteristics are produced by immersion using 5% hydrochloric acid. This is correlated with the results of other mechanical analyzes, such as the value of ash content, moisture content, pH, and viscosity. For the yield value, it has a relatively high weight when soaked in an acid solvent, namely HCl. This research method includes the preparation of cow bone samples, fat removal, mineral removal, soaking for 7 days, and extraction. The extraction process was carried out with variations times of 4 h and 6 h. The results of the study showed that the highest yield value was with 5% HCl solvent with a 4 h extraction time of 26.5% with 8.67% water content, 0.9% ash content, pH 4.64, and viscosity 3.19 cP ($p < 0.05$). A good isoelectric point is produced when using an acidic solvent, which is between 5.3–5.8. The cross-linking of gelatin with chitosan, glutaraldehyde, and glucose was successfully carried out with the FTIR absorption indicator at a wavelength of 3200 cm$^{-1}$, which indicates the presence of hydrogen bonds, 1022 cm$^{-1}$, which indicates the breakdown of aldehyde bonds in glutaraldehyde compounds into C-O bonds. According to the microbial test, when gelatin is combined with chitosan, there will be an increase in the microbial inhibition zone. This shows that the development of gelatin materials is very prospective and promising.

**Author Contributions:** Conceptualization, S.F., S.S., M.F. and B.B.; methodology, S.F., S.S., B.B.; writing—original draft, S.F., S.S., M.F. and B.B.; writing—review, S.F., S.S.; editing: S.F., B.B.; methodology, writ-ing—original draft, S.F., S.S., M.F. and B.B.; data curation: S.F., S.S., B.B.; investigation: M.F., S.F., B.B. All authors have read and agreed to the published version of the manuscript.

**Funding:** This research was funded by the Ministry of Research, Technology, Education and Culture of the Republic of Indonesia with contract number 1927/UN1/DITLIT/Dit-Lit/PT.01.03/2022. Thank you to all those who have helped to complete this research funding.

**Institutional Review Board Statement:** Not Appliable.

**Informed Consent Statement:** Not Appliable.

**Data Availability Statement:** Not Appliable.

**Conflicts of Interest:** The authors declare no conflict of interest.

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
