# Peer review of "Characterization and Development of Gelatin from Cow Bones: Investigation of the Effect of Solvents Used for Soaking Beef Bones"

_applsci, doi:10.3390/app13031550_

Round 1

Reviewer 1 Report

The authors reported the effect of different solvents (sodium hydroxide, hydrochloric acid, sulfuric acid and sodium bicarbonate) on the extraction of gelatin from cow bones. The resulted products were characterized with different methodologies such as viscosity, pH, FTIR and SEM.

-        The manuscript needs to be complemented with several basic techniques oriented to the characterization of gelatin. It includes: Bloom degree measurements, hydroxyproline content and isoelectric point. Also, I strongly recommend the amino acid sequence determination.

-        The authors must include more literature to confront the obtained results against previous works on the extraction and characterization of gelatin. Fortunately, there are plenty of those papers on the internet. The present manuscript only includes 12 references.

-        It is important to be sure if the extracted product is gelatin or hydrolyzed collagen (HC). The physical and chemical properties of both biomaterials are different. As example; HC does not form gels. Molecular weight of HC is lower than gelatin. The isoelectric point within different types of gelatins (A and B) is completely different. Extraction of gelatin under alkaline conditions produce deamidation of the protein chains, therefore, different electrostatic properties. Additionally, the authors reported a medical thread in table 1. ¿is it the effect of the collagen reassembling?

-        Figures 2-6 should include Standard deviations and statistical differences represented with letters. Then re-write the discussions including some references.

-        Finally, it is compulsory that the authors highlight the state of the art and the contribution of the paper at the introduction section.

Author Response

I would like to thank you for the suggestion for my article.

Reviewer 2 Report

The present manuscript is well-written and presented some interesting findings on the application of cow bone as raw material during gelatine production. The topic is very relevant to the present focus on by-product utilization. The language is clear and easy. However, authors should focus the abstract and discussion on the main topic, so to improve the hypothesis. 

My observations are as follows-

1.     Abstract: Line 11-14: are more general and may better fit in the Introduction rather than in the abstract. Rather I would recommend rewriting the abstract for a wide readership.

2.     Keywords: need improvement

3.     Line 52-62 needs improvement so to strengthen the hypothesis. May be more discussion on solvents, time, and cross-linkages used in the present study, rather than describing the bone structure and its composition, plz.

4.     Line 78: Authors have missed KBr full name?

5.     Line 84: Every solvent has different levels or these levels are for all ? plz clarify

6.     Line 90-91: plz rewrite the sentence

7.     I would recommend a flow chart of gelatine extraction for better understanding of the experimental design.

8.     Results and discussion: please concise first 3 para.

9.     Line 295-302: plz check whether microbial quality also conducted in the experiment, if yes then please add in the methodology section

Author Response

i would like to thank you for the suggestion for my article

Reviewer 3 Report

Dear Authors,

I studied your manuscript entitled "Characterization and Development of Gelatin from Cow Bones: Investigation of the Effect of Solvents Used for Soaking Beef Bones". Some spaces need to be improved in terms of journal quality. I recommend a major revision before further consideration for publication in the Applied Sciences.

1) The quality of the abstract and conclusion should be enhanced by the inclusion of significant research findings. More quantitative data in these sections would be beneficial.

2) The recent literature review should be summarized for benchmarking purposes and discussed in detail with your research findings.

3) How did you select the used solvents? It might also be a good idea to do an economic analysis of the solvents that were used.

4) Picture/Table captions could provide more details, so that readers will not have to go back to the experimental section for necessary information.

5) Additional analyses such as molecular weight measurements and XRD/XPS could also be presented and discussed. Please also report quantitative data obtained from EDX.

6) Please provide manufacturer details (model, city, or country) for all characterization instruments.

7) The manuscript needs to be thoroughly revised because it contains a few typos and errors.

Author Response

(The authors gave the same response as above.)

Round 2

Reviewer 1 Report

The recommendations were not attended as suggested. Please look at the recommendations carefully and answer in English language (see the authors response document)

Author Response

Thank you for the suggestion, we have made revisions according to suggestions and input. Our changes are given a blue color.

Reviewer 3 Report

Dear Author,

I studied your revised manuscript entitled "Characterization and Development of Gelatin from Cow Bones: Investigation of the Effect of Solvents Used for Soaking Beef Bones". Although you have improved its quality, some of my comments have been ignored.

1) Add a decent-sized scale bar for all SEM images.

2) The weakest component of the manuscript is the low number of references. The recent literature review should be summarized for benchmarking purposes and discussed in detail with your research findings. Some related papers were strongly suggested to be used:

a) https://doi.org/10.1016/j.matpr.2020.12.922

b) https://doi.org/10.1016/j.tifs.2017.08.012

c) https://doi.org/10.1080/00914037.2020.1740987

Author Response

(The authors gave the same response as above.)

Round 3

Reviewer 1 Report

The authors attended most of the recommendations

Author Response

Dear The reviewer 1, 

We are very grateful for the recommendation

Best regards,

authors
